# Corrosion Behavior of TiNi Alloy Fabricated by Selective Laser Melting in Simulated Saliva

**Chenfan Jia** [1], **Xinyu Wang** [2,*], **Ming Hu** [1], **Yucheng Su** [2], **Shujun Li** [3,*], **Xin Gai** [3] and **Liyuan Sheng** [4]

[1] School of Materials Science and Engineering, Jiamusi University, Jiamusi 154007, China; 15566530380@163.com (C.J.); minghu02@jmsu.edu.cn (M.H.)
[2] School of Dentistry, Jiamusi University, Jiamusi 154007, China; yuchengsu@163.com
[3] China Institute of Metal Research, Chinese Academy of Sciences, Shenyang 110016, China; xgai15s@imr.ac.cn
[4] Shenzhen Institute, Peking University, Shenzhen 518057, China; lysheng@yeah.net
[*] Correspondence: wangxinyu@jmsu.edu.cn (X.W.); shjli@imr.ac.cn (S.L.)

**Abstract:** In this work, TiNi samples were prepared by Selective Laser Melting (SLM) technology, and the influence of microstructure, fluoride ion, and pH value on corrosion behavior in a saline environment was investigated and compared with TiNi alloy fabricated by traditional forging technology. The results indicated that the corrosion resistance of the SLM sample was slightly superior to that of the wrought sample in a saline environment due to the uniform and dense oxide film formed on the SLM sample surface. However, in acidic Artificial Saliva Solution (ASS) containing fluoride ions, the corrosion current density of the SLM sample increased from $9.85 \times 10^{-2}$ to 13.9 $\mu A/cm^2$ because of the presence of $F^-$. Fluorine ions disrupted the passive film on the surface, and the Ti-F compound formed in the film, which deteriorated the corrosion resistance of the SLM sample. The increase in fluoride concentration and the decrease in pH value could accelerate the corrosion of the SLM sample.

**Keywords:** TiNi alloy; selective laser melting; corrosion resistance; fluoride ion; pH value

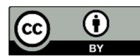

## 1. Introduction

TiNi alloy possesses excellent superelasticity, shape memory effect, biocompatibility, and damping characteristics [1–3], which have been studied extensively and found applications in the orthodontics [4,5], brackets [6], and bone implants field [7]. Along with the further development of science and technology, the application of TiNi alloy in various fields has been expanded. Up to now, alloys with shape memory effect, including Nitinol, have been used together with various types of components in automobile manufacturing, aircraft construction and household appliances, etc. [8]. Recently, selective laser melting (SLM) technology has been used to prepare TiNi alloy, which is better known by the term 4D printing. This term means the use of 3D printing technology to create objects containing various materials, which are defined as smart materials with unique or other properties that change shape over time [9]. Addictive manufacturing technology can fabricate products with a complex structure [10], high dimensional accuracy [11], and a clean surface with low impurities [12]. Thus, it has attracted extensive attention in the biomedical field. In order to obtain high-quality TiNi products, SLM process conditions or parameters must be optimized. To ensure the long-term and safe application of SLM-prepared TiNi alloy in the human body, it is necessary for it to possess excellent mechanical properties, corrosion resistance, and biocompatibility requirements [13].

Saliva is an electrolyte in the oral cavity, which can cause electrochemical corrosion of implants. The significant variation of pH value and the existence of micro-organisms in saliva can lead to the rapid dissolution of the alloy and the aggravation of corrosion, which

means the corrosion medium can influence an alloy's corrosion properties [14,15]. Dental caries prevention, for example, mouthwash and toothpaste, often contain fluoride, which can react with bacterial products to generate hydrofluoric acid (HF), resulting in the dissolution of the oxide layer on the alloy surface [16]. When TiNi is exposed to solutions containing fluoride ($F^-$), the release of metal ions increases, potentially causing oxidative stress and genotoxic damage to oral mucosa cells [17]. TiNi has excellent corrosion resistance in NaCl physiological solution; however, the presence of $F^-$ and low pH in a simulated oral environment can lead to its local and uniform corrosion [1]. In addition, no significant difference in corrosion resistance was found between strained and unstrained TiNi. In a fluorine-containing medium, TiNi has the risk of corrosion-induced brittle fracture, and various physiological environments and production processes can affect its corrosion behavior [18].

Currently, there are extensive studies on the corrosion behavior of TiNi alloy prepared by traditional technology. However, few have focused on the corrosion characteristics of 3D-printed TiNi in an oral environment. In this work, TiNi alloy was prepared by SLM technology, and the corrosion behavior of SLM alloy in an acidic solution containing fluorine was evaluated and compared with wrought TiNi.

## 2. Materials and Methods

### 2.1. Materials

The TiNi parts with dimensions of 10 mm × 10 mm × 10 mm were fabricated using a Realizer SLM 250 machine (Realizer GmbH, Borchen, Germany). TiNi pre-alloyed powder with particle sizes ranging from 15 to 60 μm was melted by the selective laser. The laser power was 190 W, the diameter of the focused spot was 40 μm, the thickness of the fixed powder layer and scanning interval was 30 μm and 60 μm, and the scanning rate was 900 mm/s. Wrought TiNi samples were provided by IMR.

### 2.2. Microstructure and Characterization

Microstructural characterization was performed by optical microscopy (OM) (ZEISS-AXIO, Jena, Germany), scanning electron microscopy SEM (JSM-6510A, JEOL, Musashino, Japan), and Energy Dispersive Spectrometer (EDS, JEOL, Musashino, Japan). The specimens for the OM analysis were mechanically polished and then etched in the corrosive liquid consisting of 10 vol. % HF, 40 vol. % $HNO_3$, and 50 vol. % $H_2O$. The corrosion morphology and element distribution of the SLM and wrought samples were observed by SEM and EDS. Phase compositions were examined by X-ray diffraction (XRD, D/max 2400 diffractometer, Rigaku, Akishima, Japan ) using a Cu-K$\alpha$ radiation source. Analysis of XRD data was in conformity with the corresponding JCPDS No. 18-0899, 35-1281 and 39-1113 for B2, B19′ and $Ti_3Ni_4$ phases using Jade 5.0 software.

### 2.3. Electrochemical Measurements

Electrical spark cutting was used to cut SLM and wrought TiNi samples to obtain a working surface of a cross-sectional area of 1.0 cm². The back surface of the samples was connected with copper wire by argon arc welding. Then, the samples were sealed in plastic tubes with epoxy resin. The sample surfaces were polished with 2000# with SiC sandpaper and cleaned with anhydrous ethanol and water for 5 min, respectively, and blown dry.

The electrochemical measurements were carried out in ASS composed of (g/L) 0.4 NaCl, 0.4 KCl, 0.69 $NaH_2PO_4 \cdot 2H_2O$, 0.906 $CaCl_2 \cdot 2H_2O$, 0.005 $Na_2S \cdot 9H_2O$, and 1.0 $CH_4N_2O$. The pH was adjusted to 4 and 5 using a 2% lactic acid solution. Then, adding NaF into the corrosive medium, the corrosion behavior of the TiNi samples in ASS containing 0%, 0.2%, and 0.5%$F^-$ and with pH values of 4 and 5 were measured.

Electrochemical corrosion test data were collected by a computer-controlled potentiostat (PARSTAT 2273, AMETEK, San Diego, CA, USA) using a three-electrode system,

with the platinum electrode as an auxiliary electrode (CE), saturated calomel (SCE) as a reference electrode (RE), and the working electrode (specimen). The experiments were carried out in a water bath maintained at 37 ± 1 °C. After pouring 300 mL of fresh electrolyte into the electrolytic cell, the open-circuit potential (OCP) tests were recorded for 4500 s. The electrochemical impedance spectroscopy (EIS) testing in the frequency range of $10^5$–$10^{-2}$ Hz was conducted at the open-circuit potential. The spectra were interpreted utilizing ZsimpWin 3.10 software (EG&G, Montréal, QC, Canada) as well as fitted to obtain a suitable equivalent circuit. Subsequent dynamic potential polarization measurements were performed at a scan rate of 0.333 mV/s from −500 mV with respect to OCP to 1600 mV.

### 3. Results

#### 3.1. Microstructure and Phase Composition

Figure 1 shows the XRD patterns of the wrought and SLM TiNi samples. Both samples were primarily composed of the B2 austenite phase. The wrought sample contained a small amount of $Ti_3Ni_4$ phase because of the slow precipitation behavior in the Ni-rich TiNi alloy [19]. Compared to the wrought sample, it is found that the intensity of B2 (200) became higher and broader, and a small amount of B19′ martensite phase existed in the SLM sample due to the rapid solidification and repeated heating during SLM fabrication [11].

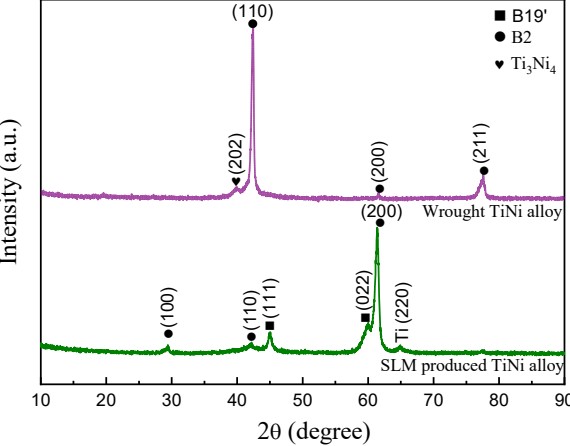

**Figure 1.** XRD patterns of wrought and SLM TiNi alloy.

The microstructure image of wrought and SLM TiNi samples is shown in Figure 2. It is seen that the SLM sample exhibits a uniform equiaxed grain with a size of approximately 132.64 ± 18.01 μm. In the wrought samples, the equiaxed grain size is around 51.01 ± 9.93 μm. There were many inclusions precipitated in the grains and along the grain boundaries (Figure 2b).

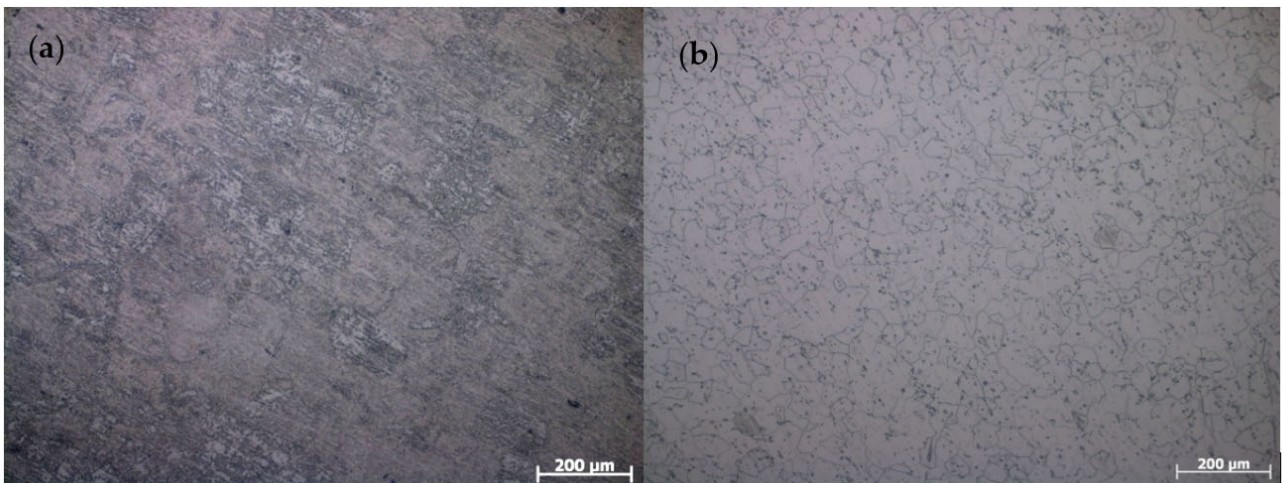

**Figure 2.** Microstructure of SLM alloy and wrought TiNi. (**a**) SLM alloy; (**b**) Wrought TiNi.

*3.2. OCP of TiNi Alloy in the ASS*

3.2.1. OCP of SLM and Wrought TiNi Sample

As can be seen, Figure 3 shows the open circuit potential as a function of the time of SLM and wrought TiNi samples in ASS with a pH value of 4 and 0.5% F$^-$, respectively. The abrupt positive displacement of $E_{ocp}$ suggested a protective oxide film formation on the surface of the TiNi sample. The oxide layer was unstable after contact with the corrosive medium and was in a dynamic balance between film formation and dissolution [20]. Following the immersion of the wrought sample in ASS, there was a significant potential fluctuation in the OCP curve. $E_{ocp}$ decreased gradually and reached −0.39 V at the end of the test. Similar to the wrought sample, the $E_{ocp}$ of the SLM sample showed a downward trend. After soaking in ASS for 4500 S, $E_{ocp}$ reached 0.38 V, slightly higher than that of wrought samples. The corrosion trend of the two studied samples in simulated saliva containing F$^-$ was similar. However, compared with the wrought sample, the SLM sample exhibited a slightly higher $E_{ocp}$ value.

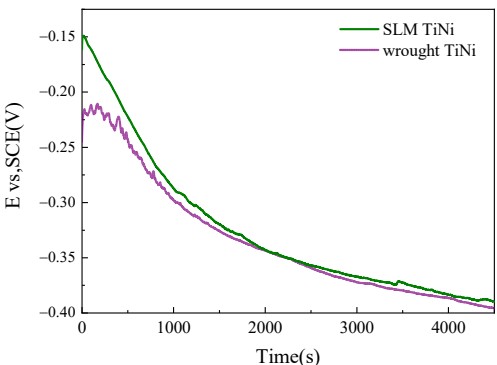

**Figure 3.** OCP diagram of SLM and wrought TiNi samples in ASS with a pH value of 4 and 0.5% F$^-$.

3.2.2. OCP of SLM TiNi Sample in ASS with Different pH and F$^-$ Concentration

The OCP diagram of the SLM TiNi samples with different pH and F$^-$ values in 37 ± 1 °C ASS is shown in Figure 4. When there was no F$^-$ in ASS, the $E_{ocp}$ of the SLM sample was approximately −0.25 V, and the $E_{ocp}$ decreased to −0.35 V after F$^-$ addition. With the increase in F$^-$ concentration (0.5%), the $E_{ocp}$ of the SLM sample decreased. Under the same

F$^-$ concentration, the $E_{ocp}$ of the sample decreased with pH value. The above results indicated that the increase of F$^-$ and the decrease of pH value enhance the corrosion tendency of SLM samples.

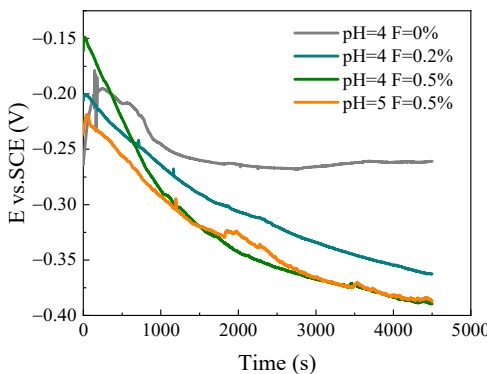

**Figure 4.** OCP diagram of the SLM alloy in ASS with different pH and F$^-$ concentrations.

### 3.3. EIS of TiNi Alloy in ASS

### 3.3.1. EIS of SLM and Wrought TiNi Samples

The electrochemical impedance spectra of SLM and wrought TiNi samples in ASS with a pH value of 4 and 0.5% F$^-$ at the $E_{ocp}$ are presented as Nyquist diagrams and Bode plots (Figure 5). For the SLM sample, the Nyquist diagrams (Figure 5a) exhibit one capacitance loop. It is seen from the figures that the relationship between the real part Z′ and the imaginary part Z″ followed the equation:

$$\left(Z_{Re} - \frac{R_p}{2}\right)^2 + Z_{Im}^2 = \left(\frac{R_p}{2}\right)^2 \tag{1}$$

where $R_P$ is the polarization resistance. The larger the $R_P$, the larger the arc radius of capacitive reactance and the more difficult the disruption of the passive film. From Figure 5a, both SLM and wrought samples had a single capacitive arc and showed a similar radius.

Its Bode magnitude plot is characterized by three distinct regions. In the high frequency ($10^3$–$10^5$ Hz), a horizontal straight line independent of frequency change (slope ≈ 0) is observed due to the response of electrolyte resistance. In the middle-frequency ranges ($10^0$–$10^3$ Hz), the impedance spectrum displays a linear relation (with a slope of approximately −1) between log |Impedance| and log (Frequency), a typical capacitive behavior. Furthermore, the modulus of impedance at low frequency reaches the magnitude order of $10^4$ Ω·cm². From the Bode phase diagram, it can be seen that the phase angle decreases to zero at high frequency and slightly decreases to a lower value at low frequency. The former shows that the impedance in the high-frequency region is mainly determined by the solution resistance, while the latter shows the contribution of passivation film resistance to impedance. The phase angle maximum is about 70°, emerging in the middle frequency ranges.

For the wrought sample, the capacitance loop with the smaller radius in Nyquist diagrams can be observed in Figure 5a. Many overlapping parts in the Bode diagram of the studied samples show that they present almost the same corrosion resistance. The maximum phase angle of both samples appeared in the range of intermediate frequency. It means that a single passivation film formed on the surface of both samples.

As a physical model of electrochemical reaction on the electrode, an equivalent circuit can be used to interpret electrochemical corrosion behavior from EIS. The existence of a compact oxide film on SLM and the wrought sample surface is represented by an equivalent circuit with one time constant. The $R_s$ ($Q_bR_b$) system can better simulate the system's

corrosion process (Figure 6). $R_s$ represents the solution resistance, $Q_b$ denotes the non-Faraday impedance, and $R_b$ refers to the Faraday impedance. Due to electrode surface roughness and energy dissipation, the actual capacitance deviates from the ideal capacitance. Therefore, it is proposed that the Constant Phase Angle Element (CPE) can represent the deviation of the ideal capacitor. The impedance of CPE is defined as:

$$Z = \frac{1}{Y_0}(jw)^{-n} \tag{2}$$

where n is related to the surface roughness, surface defects, and uniformity of samples. Generally, $0 < n < 1$, n = 0 is equivalent to pure resistance, and n = 1 is equivalent to pure capacitance.

　　Table 1 summarises the fitting results from the EIS data shown in Figure 5. Experimental errors are all $10^{-3}$ orders of magnitude, and fitting results are in good agreement with experimental results. As seen in Table 1, the solution resistance ($R_s$) values are close for both samples in ASS. $R_b$ reflects the resistance of film formed on the metal surface; the larger the $R_b$, the higher the resistance to F⁻ corrosion in an acidic solution. Accordingly, the $R_b$ (2.585 ± 0.594 KΩ·cm²) of the SLM TiNi surface passivation film had little difference from that of wrought TiNi (2.484 ± 0.223 KΩ·cm²). Therefore, the passivation film that formed on the SLM sample had the same quality as that formed on the wrought sample.

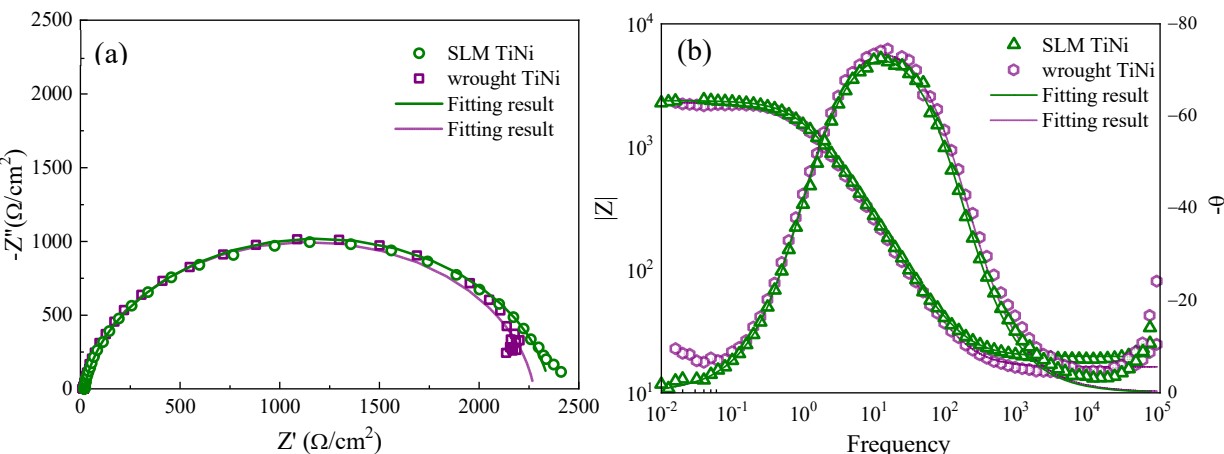

**Figure 5.** EIS of SLM and wrought TiNi samples at ASS with a pH value of 4 and 0.5%F⁻; (**a**) Nyquist; (**b**) Bode.

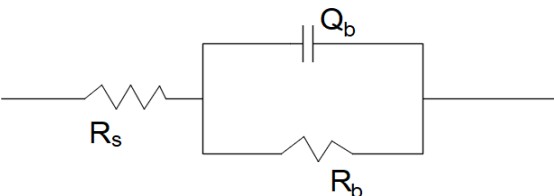

**Figure 6.** Equivalent circuit diagram of SLM and wrought TiNi sample in ASS.

**Table 1.** EIS fitting parameters of SLM and wrought TiNi sample in ASS.

| Samples | $R_s/\Omega\cdot cm^2$ | $R_b/K\cdot\Omega cm^2$ | $Q_b/\mu F cm^{-2}$ | $n_b$ | Chi-Square |
|---------|-----------|-----------|-----------|-------|------------|
| SLM TiNi | $22.24 \pm 1.98$ | $2.585 \pm 0.594$ | $82.49 \pm 0.67$ | $0.9189 \pm 0.0052$ | $(5.89 \pm 0.75) \times 10^{-3}$ |
| Wrought TiNi | $18.75 \pm 2.07$ | $2.484 \pm 0.223$ | $74.36 \pm 7.71$ | $0.9205 \pm 0.0066$ | $(4.81 \pm 0.45) \times 10^{-3}$ |

3.3.2. EIS of SLM TiNi Sample in ASS with Different pH and $F^-$ Concentrations

The EIS measurements of the SLM TiNi samples were carried out in ASS with different pH and $F^-$ concentrations, including Nyquist diagrams and Bode plots (Figure 7). The Nyquist diagram shows a single capacitive loop with a larger radius in ASS solution at pH 4 without fluoride ions. When $F^-$ was added (0.2%), the radius was reduced, indicating that the existence of $F^-$ had a significant influence on the sample surface. With the increase of $F^-$ concentration to 0.5%, the smaller capacitive loop observed indicates the increase in $F^-$ concentration causes the decrease of TiNi surface impedance. by adjusting the pH of ASS with $F^-$ concentration of 0.5% to 5, a larger capacitive loop was observed compared with that with a pH of 4. Therefore, the decrease in the pH value of the medium hinders the passive membrane electrode reaction.

As can be seen from the Bode diagram, a significant difference existed between the presence or absence of $F^-$ in the medium in the low-frequency region. When $F^-$ did not exist, its Bode magnitude was characterized by two distinct regions; the low and middle-frequency region was a straight line with a slope of about −1. In addition, |Z| at low frequency reaches the magnitude order of $10^6$ $\Omega\cdot cm^2$. After adding $F^-$, the curve in the low-frequency region ($10^{-1}$–$10^{-2}$) became close to a horizontal line. Along with the fluoride concentration, the curve is approximately horizontal in the frequency range of $10^0$–$10^{-1}$, and |Z| is reduced to $10^4$ order of magnitude. The Bode diagram of SLM alloy in ASS solution with pH values between 4 and 5 is almost overlapped except for in the frequency range of $10^0$–$10^{-2}$, performing the slightly reduced |Z| at low frequency at pH 4 in comparison to that of pH 5. The maximum phase angle appeared in the intermediate frequency region. When there was no $F^-$ in the solution, the maximum phase angle exceeded 80°, as evidenced by the plateaus at frequencies of about 0.1 to 10 Hz. When (0.2%) $F^-$ was added to the solution, the maximum phase angle and its corresponding frequency range were reduced, indicating a denser passivation film on the surface without $F^-$. With increasing $F^-$ concentration (0.5%), the maximum phase angle and the corresponding frequency range slightly decreased.

Compared with the curve in 0.5% $F^-$ solution at pH 4, the curves at pH 5 were observed that the maximum phase angle and its corresponding frequency range slightly increased, which means the stability of the passivation film increased. According to the characteristics of the impedance spectra of the SLM sample under four corrosion conditions, the same equivalent circuit model $R_s$ ($Q_b R_b$) (Figure 6) is used to describe the electrode reaction. Table 2 provides the obtained electrochemical parameters from the EIS data shown in Figure 7. The existence and increase of $F^-$ concentration significantly influenced the corrosion behavior of the passivation film on the SLM alloy's surface, making its corrosion resistance worse. Furthermore, $R_b$ slightly increased when the pH value of the medium increased, signifying that the increase in $H^+$ concentration could promote the corrosion reaction of the SLM samples. As such, from the viewpoint and fitting results of EIS measurement, high concentration of fluoride and low pH value make SLM sample possesses fragile oxide films.

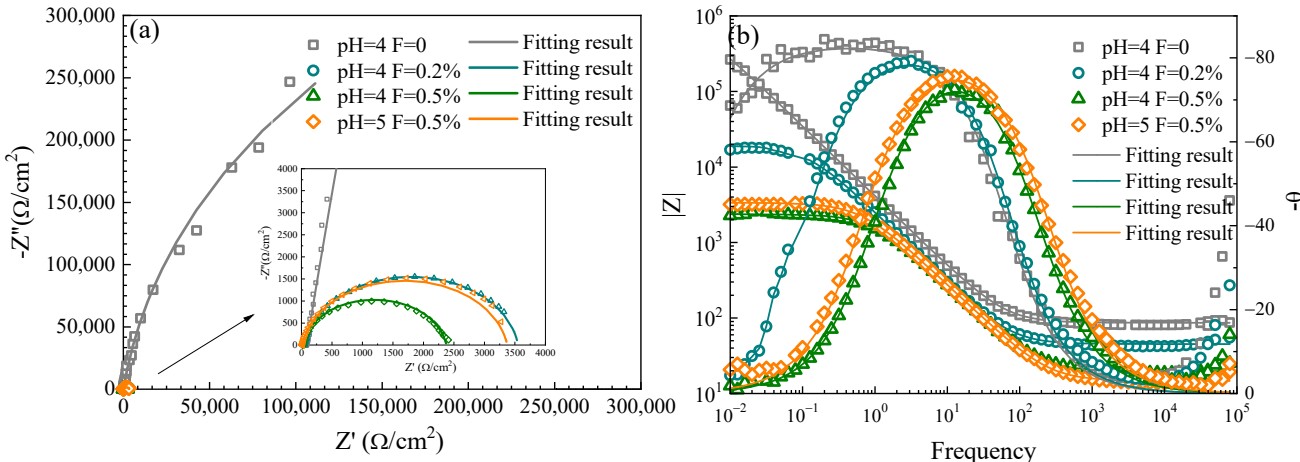

**Figure 7.** EIS diagram of SLM sample in ASS with different pH values and F$^-$ concentrations. (**a**) Nyquist; (**b**) Bode.

**Table 2.** EIS fitting parameters of SLM and wrought TiNi sample in ASS.

| Samples | $R_s$/$\Omega$·cm² | $R_b$/K$\Omega$·cm² | $Q_b$/$\mu$Fcm$^{-2}$ | $n_b$ | Chi-Square |
|---|---|---|---|---|---|
| pH 4 without F$^-$ | 90.09 ± 2.40 | 542.967 ± 72.206 | 45.31 ± 7.25 | 0.9267 ± 0.0011 | (2.85 ± 0.43) × 10$^{-2}$ |
| pH 4 0.2% F$^-$ | 47.51 ± 4.30 | 4.237 ± 0.653 | 74.31 ± 2.16 | 0.9246 ± 0.0013 | (1.44 ± 0.12) × 10$^{-2}$ |
| pH 4 0.5% F$^-$ | 22.24 ± 1.98 | 2.585 ± 0.594 | 82.49 ± 0.67 | 0.9189 ± 0.0052 | (5.89 ± 0.75) × 10$^{-3}$ |
| pH 5 0.5% F$^-$ | 18.12 ± 2.60 | 3.674 ± 0.231 | 80.42 ± 4.52 | 0.9203 ± 0.0104 | (6.31 ± 1.15) × 10$^{-3}$ |

*3.4. Potentiodynamic Polarization Curve of TiNi Alloy in ASS*

3.4.1. Polarization Curves of SLM Alloy and Wrought TiNi

At pH 4 and 0.5% F$^-$ the potentiodynamic polarization results were composed of SLM and wrought TiNi samples, without distinct differences between the two cathodic polarization curves, as shown in Figure 8. On the anodic branches, for the SLM sample, the current density increases first with increasing potential from OCP to −0.09 V. In the range of −0.09–1.13 V, the current density of the SLM sample gradually changed with voltage; when the potential reached 1.13 V, the current density increased promptly. This means there was no passivation interval with a small variation range of current density.

The nature of anodic polarization curves for the wrought sample reveals a similar stable passive behavior, except for the rapid increase of current density at 0.42 V. The anodic and cathodic polarization curves represent the passivation and oxygen absorption of the sample, respectively. The existence of the SLM sample in ASS solution, the dissolution reaction, and the beginning of the passivation reaction will not produce a clear experimental anode Tafel region in the anodic polarization curve of the SLM and wrought TiNi samples in ASS.

According to the Tafel extrapolation method, self-corrosion potential ($E_{corr}$) and self-corrosion current density ($I_{corr}$) were determined, as shown in Table 3. The $E_{corr}$ for the SLM sample was estimated to be −0.38 ± 0.006 V, whereas the $E_{corr}$ value for the wrought sample was −0.41 ± 0.009 V. The $E_{corr}$ of the SLM sample was slightly higher than that of the wrought sample, indicating that the wrought sample was prone to corrosion. The corrosion current density ($I_{corr}$) that represents the corrosion rate for the SLM sample is 22.0 ±

0.13 μA/cm², which is slightly higher than that of the wrought sample (11.8 ± 1.57 μA/cm²). When the potential continued to increase, the range of current density of the SLM sample with voltage changes was smaller, while the corrosion rate of the wrought sample significantly increased. It was noticed that the $E_{corr}$ and $I_{corr}$ values for both samples were very close, while the breakthrough potential ($E_{pit}$) for the SLM sample was apparently greater than that of the wrought sample. This indicates that the stability of passive film forming on the SLM sample is slightly superior to that on the wrought sample.

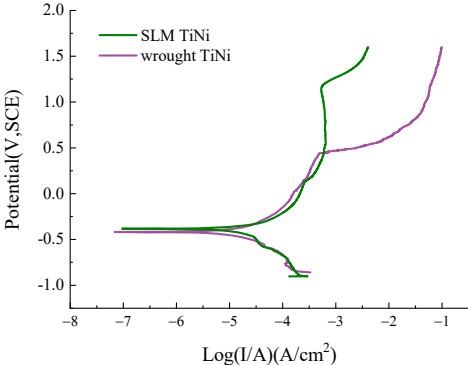

**Figure 8.** ASS polarization curves of SLM and wrought TiNi samples in 0.5% F⁻ ASS at pH 4.

**Table 3.** Potentiodynamic polarization corrosion parameters of SLM and wrought TiNi samples in 0.5% F⁻ ASS at pH 4.

| Corrosion parameters | SLM TiNi | Wrought TiNi |
|:---:|:---:|:---:|
| $E_{corr}$ (V) | −0.38 ± 0.006 | −0.41 ± 0.009 |
| $I_{corr}$ (μA/cm²) | 22.0 ± 0.13 | 11.8 ± 1.57 |
| $I_{pass}$ (μA/cm²) | 181 ± 9.18 | 55.46 ± 7.11 |
| $E_{pit}$ (V) | 1.13 ± 0.02 | 0.42 ± 0.02 |

3.4.2. Polarization Curves of SLM TiNi Samples in ASS with Different pH and F⁻ Concentrations

Figure 9 show the potentiodynamic results of SLM TiNi samples in ASS with different pH and F⁻ concentrations at 37 °C. It can be seen that the SLM sample in ASS solution with different pH and fluoride concentrations has the same cathodic reaction with different rates. For better visualization of the polarization behavior of the SLM sample in ASS, the corrosion parameters, including $E_{corr}$, $I_{corr}$, $I_{pass}$, and $E_{pit}$, are listed in Table 4 for comparison.

In pure (fluorine-free) ASS electrolyte at pH 4, the $E_{corr}$ was −0.30 V, and the $I_{corr}$ was $9.85 \times 10^{-2}$ μA/cm². After the voltage was greater than 0.10 V, an obvious passivation area was observed on the anodic polarization curve. The evaluation of the passivation ability of TiNi alloy mainly depends upon two aspects: firstly, the $E_{pp}$ (initial passivation potential) and $I_{pass}$ (passivation current density) are two indexes that represent the ease of the material entry into the passivation stage; secondly, whether the passivation state is stable depends on $I_{pp}$ (passive current density) and the passivation interval, that is, the difference between breakdown potential ($E_{pit}$) and $E_{pp}$. $I_{pp}$ means the current density in which the passivation films on the surface of metal remain stable and protective. A small $I_{pp}$ value and a larger interval lead to higher stability of the passivation state. The passivation current density ($I_{pass}$) was 1.25 ± 0.16 μA/cm², and $I_{pp}$ fluctuated around 1.31 ± 0.37 μA/cm² during passivation. When the potential increased and entered the dissolution zone, the ($E_{pit}$) was 1.27 V, while the passivation interval was 1.17 V.

A different corrosion resistance behavior was observed in the solution containing fluoride, whereas no passivation behavior was found in the SLM sample. In 0.2% F⁻ ASS solution at pH 4.0, the existence of F⁻ enabled a negative shift of $E_{corr}$ to −0.36 V and the increase of $I_{corr}$ (13.9 μA/cm²) by two magnitude orders. Entering the passivation zone at −0.23 V, the $I_{pass}$ (34.8 ± 0.51 μA/cm²) increased by an order of magnitude compared to that of fluorine-free ASS solutions. When the potential reached $E_{pit}$ (1.14 V), the current density of the SLM sample rapidly increased. The current density changed slightly in the −0.23–1.14 V range. Fluoride ion was very negatively and significantly correlated with the electrochemical corrosion behavior of the SLM sample. Potentiostatic tests were carried out on the SLM sample in 0.5% fluorine ASS at pH 4; $E_{corr}$ decreased only by 0.02 V, and its $I_{corr}$ increased by 8.1 μA/cm². In the range of −0.09–1.13 V, the current density changed very gradually with the voltage, ranging from 181–555 μA/cm². The existence of fluoride and the increase in its concentration resulted in the polarization curve of the SLM sample shifting to a lower value, implying that this condition could promote corrosion behavior.

Keeping the F⁻ concentration of the medium constant while reducing the pH value, the corrosion performance became worse. In a fluorine-containing ASS solution at pH 5, its $E_{corr}$ was −0.38 V, the same as that of pH 4. As such, there was no significant difference in corrosion potential between the fluorine-containing ASS solution at pH 4 and 5, except that its $I_{corr}$ reduced to 20.8 μA/cm², around 1.2 μA/cm² lower than that at pH 4. The change rate of current density was low at a range of −0.15–1.13 V and fluctuated at a range of 80.1–468 μA/cm², while the fluctuation range of the current density was slightly smaller. From the polarization curve measurement, the increase in F⁻ concentration and low pH in ASS were found to worsen the corrosion resistance of the SLM sample.

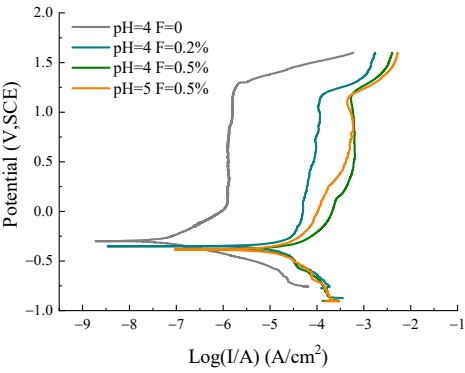

**Figure 9.** SLM TiNi sample polarization curve in ASS with different pH and F⁻ concentration.

**Table 4.** Potentiodynamic polarization corrosion parameters of SLM and wrought TiNi samples in 0.5% F⁻ ASS at pH 4.

| Corrosion parameters | pH 4 without F⁻ | pH 4 0.2% F⁻ | pH 4 0.5% F⁻ | pH 5 0.5% F⁻ |
|---|---|---|---|---|
| $E_{corr}$ (V) | −0.30 ± 0.005 | −0.36 ± 0.009 | −0.38 ± 0.006 | −0.38 ± 0.014 |
| $I_{corr}$ (μA/cm²) | (9.85 ± 0.46) × 10⁻² | 13.9 ± 3.05 | 22.0 ± 0.13 | 20.8 ± 5.85 |
| $I_{pass}$ (μA/cm²) | 1.25 ± 0.16 | 34.8 ± 0.51 | 181 ± 9.18 | 80.1 ± 5.63 |
| $E_{pit}$ (V) | 1.27 ± 0.04 | 1.14 ± 0.00 | 1.13 ± 0.02 | 1.13 ± 0.02 |

### 3.5. Corrosion Morphology of TiNi Samples in ASS

3.5.1. Corrosion Morphology of SLM and Wrought TiNi Samples

Figure 10 shows the corrosion morphology of the SLM samples after potentiodynamic polarization in 0.5% F⁻ ASS at pH 4. As seen from Figure 10a, the SLM sample formed dense corrosion pits with diameters ranging from 2.15 to 6.73 μm under this condition; Figure 10b illustrates the enlarged corrosion pits. The area in Figure 10b was analyzed by EDS, as shown in Figure 10c,d. Accordingly, the Ni element was less distributed in the pits, and its low concentration on the corroded surface signified that pitting corrosion caused the Ni ions to be released from the matrix into the corrosive medium. Meanwhile, the remaining titanium reacted with dissolved oxygen in the solution to produce titanium oxide in the corroded area [21].

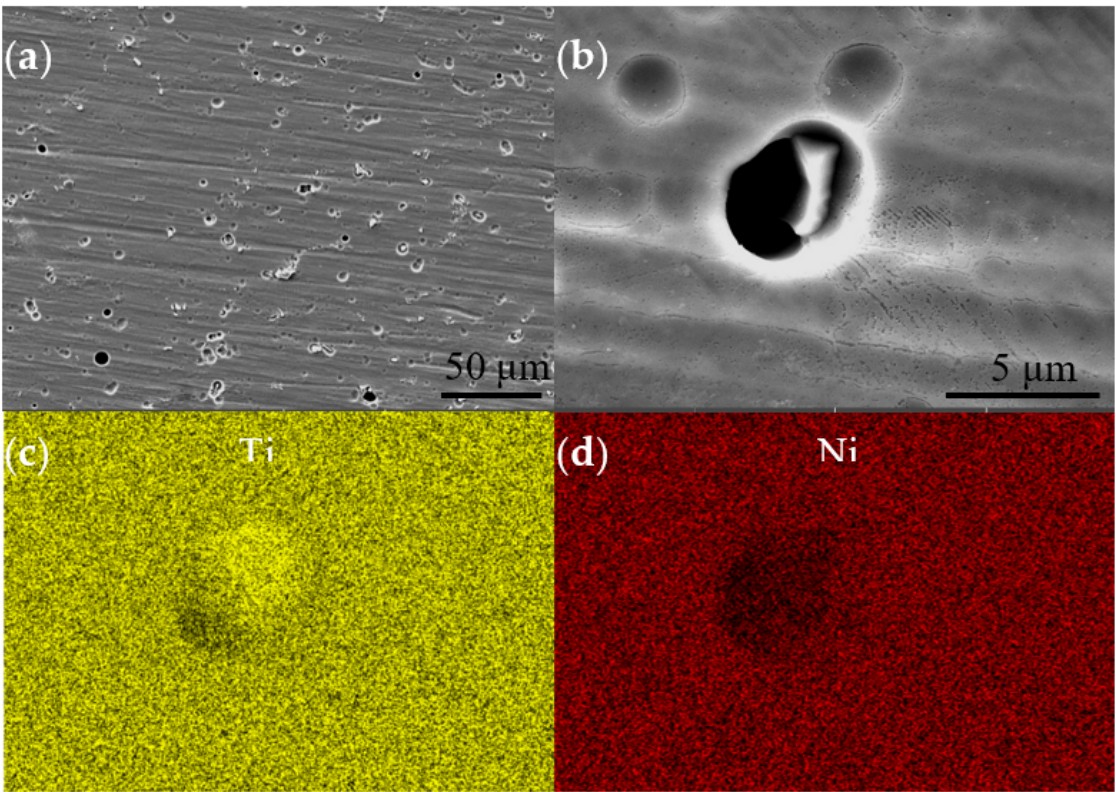

**Figure 10.** SEM and EDS images of SLM sample after corrosion in ASS in 0.5% F⁻ ASS at pH 4. (**a**) is SEM image of corrosion pits; (**b**) is enlarged SEM image of a corrosion pit shown in (a); (**c**) is EDS image of Ti for the whole area shown in (b) ; (**d**) is EDS image of Ni for the whole area shown in (b) .

Comparing SEM images (Figure 11), the corrosion morphology of the wrought sample in 0.5% F⁻ ASS at pH 4 shows a large-scale pit. Its diameters are about 1207.369 ± 65.99 μm. There were many small-scale pits around the large pits, as shown in Figure 11c, with sizes ranging from 3.04–12.02 μm. Comparing the corrosion pits of the two samples, those of wrought samples were slightly larger and deeper. EDS analysis of the etched holes indicated that the Ni content inside the etched holes was reduced, as shown in Figure 11b,d. By comparing the distribution of Ni elements in the two areas, the Ni ion concentration in the large-sized etched hole area was 35.8%, and that in the area near the etched hole area

was 50.1%. The Ni ion content in the etched hole significantly reduced after etching, consistent with the results observed in the SLM sample. Hence, it was deduced that more Ni ions release into the solution with corrosion behavior compared to Ti ions.

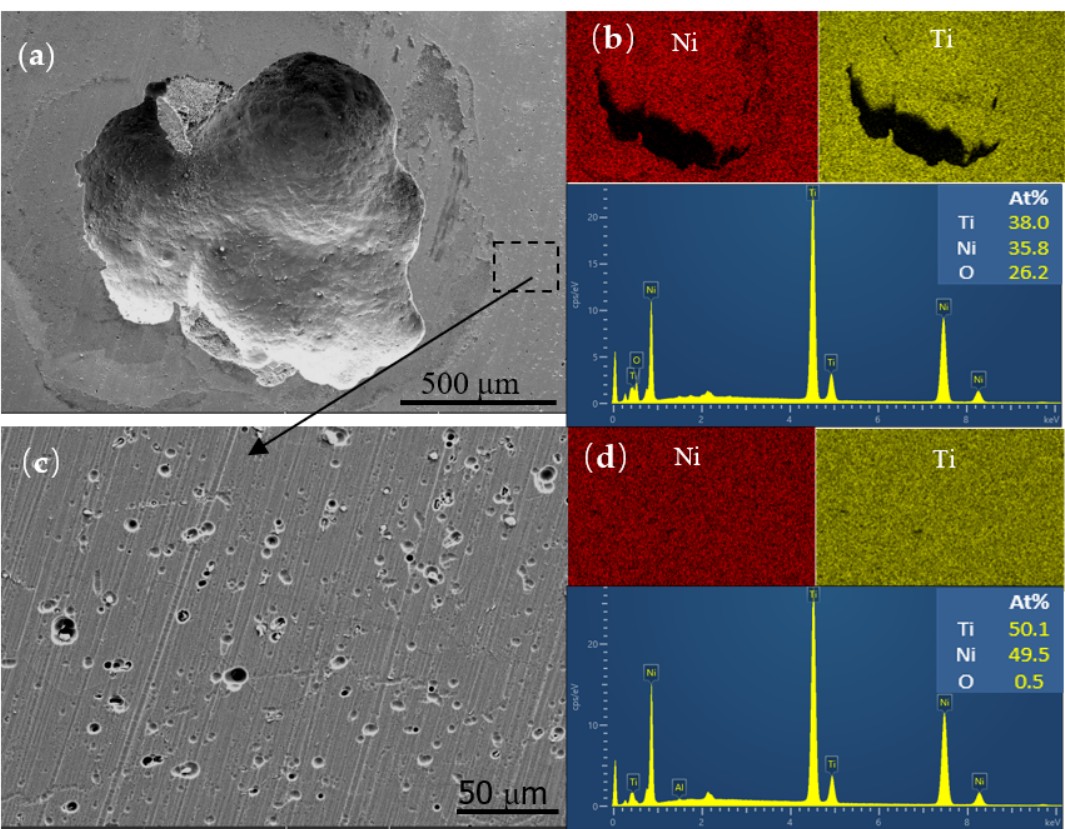

**Figure 11.** SEM and EDS images of wrought TiNi sample after corrosion in 0.5% F⁻ ASS at pH 4. (**a**) is SEM image of a large corrosion pit; (**b**) is EDS results and element distribution for the whole area shown in (a) ; (**c**) is enlarged images in the vicinity of the large corrosion pit circled in (a) ; (**d**) is EDS results and element distribution for the whole area shown in (c) .

### 3.5.2. Corrosion Morphology of SLM TiNi Samples in ASS with Different pH and F⁻ Concentrations

SLM samples exhibit different corrosion behaviors in 37 ± 1 °C ASS solution with different concentrations of fluoride ions and pH values (Figure 12). When there was no F⁻ in the solution at pH 4, a few corrosion pits could be seen in Figure 12a, which were sparse and small, with diameters ranging from 0.28–0.80 μm. After adding 0.2% F⁻ into ASS, from Figure 12b, the number of pits increased, and the size slightly increased in the range of 1.1–2.8 μm. It indicated that the existence of F⁻ evidently weakened the corrosion resistance of TiNi alloy, leading to local corrosion. Figure 12c illustrates the corrosion morphology of the sample in the solution containing 0.5% F⁻, which was relative to that of the sample in 0.2% F⁻ solution. The diameter of corrosion pits increased to about 5.5 μm, and the number of corrosion pits increased. With the increase of pH value to 5, the corrosion morphology is shown in Figure 12d. There was no obvious difference between the size of corrosion pits and pH 4, but the number of corrosion pits was reduced.

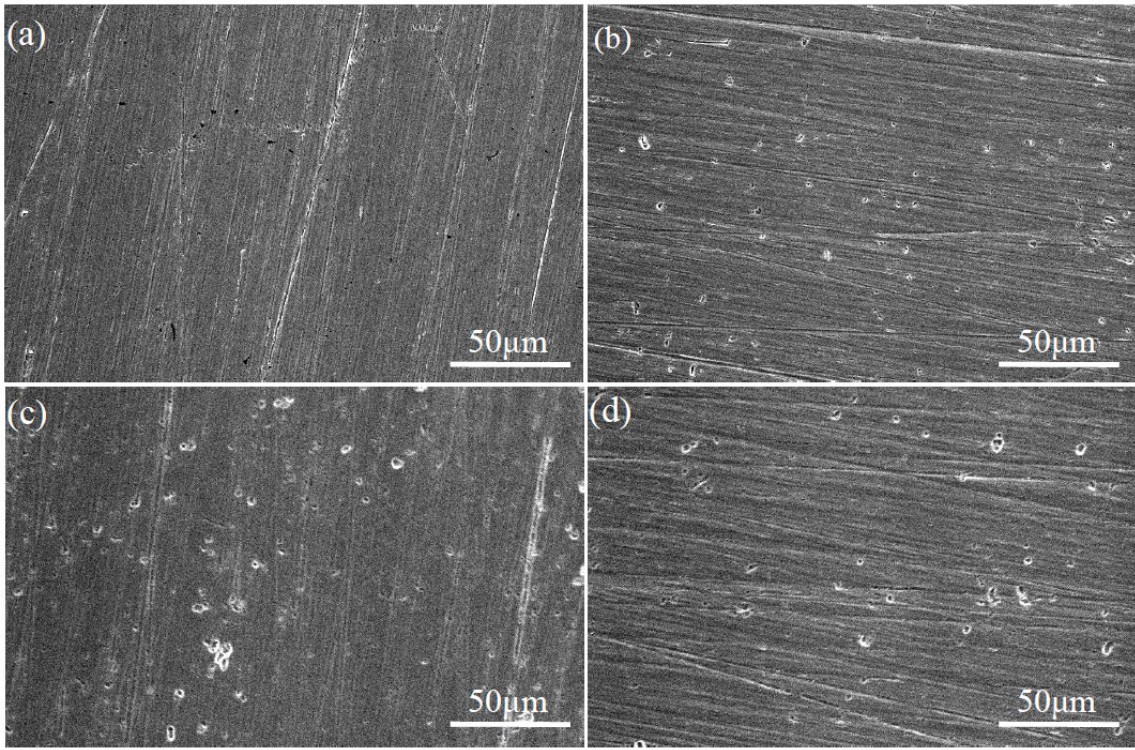

**Figure 12.** The corrosion morphology of SLM TiNi samples in ASS with different pH and F⁻ concentration. (**a**) pH 4 without F⁻; (**b**) pH 4 0.2% F⁻; (**c**) pH 4 0.5% F⁻; (**d**) pH 5 0.5% F⁻.

## 4. Discussion

### 4.1. Effect of F⁻ on Corrosion Resistance of SLM TiNi Alloy

TiNi alloy is a typical self-passivating metal. The corrosion behavior of TiNi alloy is influenced by different process parameters of electrolyte and SLM technology. Qiu [22] et al. carried out electrochemical corrosion tests on TiNi produced by SLM in simulated body fluids (PBS) and showed a dimensional passivation current density of around 3 $\mu A/cm^2$, which differs very little from the $I_{PP}$ (1.25 ± 0.16) $\mu A/cm^2$ of the SLM alloy in this thesis in ASS solution at pH 4 without fluoride ions. In addition, the SLM alloy in this work observed a wider potential interval (0.05~1.27 V) compared to that of previous studies (0.00~0.27 V). Yu [23] and others produced TiNi alloys using the SLM technique and conducted electrochemical corrosion tests in 3.5 wt% NaCl solution. The results showed that the samples prepared at low scanning speed (LP) had better corrosion resistance in 3.5 wt% NaCl solution. The $E_{corr}$ of LP-TiNi was −0.32 V, and $I_{corr}$ was 10.9 $\mu A/cm^2$. The $E_{corr}$ of SLM alloy in this paper was−0.30 V, and $I_{corr}$ was 9.85 × 10⁻² $\mu A/cm^2$ in ASS solution with pH 4.

It is known that the passivation of an alloy is closely related to the potential value, surface roughness, oxidation degree, pH value, and temperature of medium [24]. The existence of F⁻ in an acidic environment may lead to the rupture of this layer [25,26]. Nakagawa M. et al. concluded that when the HF concentration in the solution was higher than approximately 30 ppm, the passivation film that formed on CP titanium alloy was destroyed to produce the process of strong localized corrosion [27,28]. Kassab et al. found that TiNi alloy would corrode locally in fluorine-containing solutions [1]. Similar to pure Ti and other titanium alloys, the corrosion behavior of TiNi alloy depends on fluoride concentration. While F⁻ will not hinder oxide layer formation on the electrode surface, it may influence the properties of the oxide layer. Moreover, it may interfere with $TiO_2$ formation, leading to changes in the passivation layer and making it porous, and enabling contact between the metal and the electrolyte [29,30].

In this work, the addition of $F^-$ has a significant effect on the corrosion resistance of SLM alloy. In $F^-$-free solution, OCP tends to be stable; otherwise, OCP always moves in the negative direction. This result indicates that the damage of the passivation film on the TiNi alloy surface by HF in the solution cannot be promptly repaired and that TiNi cannot enter the passivation state at this time and is constantly corroded. Consequently, the number and size of corrosion pits increase with $F^-$. Comparing the corrosion behavior of TiNi alloy at pH 4 and 5, the OCP of the SLM sample was lower or higher with the decrease of pH value and more corrosion pits formed on the surface of the studied sample. In conjunction with previous studies, the corrosion resistance of TiNi alloy was better when the pH value of simulated saliva was higher in the presence of $F^-$. The pitting corrosion of oxide film on the TiNi surface largely depends on the chemical properties and the concentration of anions in the solution. Additionally, there was a strong interaction between $Ti^{4+}$ and $F^-$ at the oxide film/electrolyte interface. A water-soluble Ti-F complex can be formed and lead to the dissolution of the sample in the presence of HF and fluoride [27, 31–34]. The $TiO_2$ layer can react with $F^-$ in several ways according to the followed equations:

$$F^- + H^+ \rightarrow HF$$

$$Ti_2O_3 + 6HF \rightarrow 2TiF_3 + 3H_2O$$

$$TiO_2 + 4HF \rightarrow TiF_4 + 2H_2O$$

$$TiO_2 + 2HF \rightarrow TiOF_2 + H_2O$$

$$TiO_2 + 6F^- + 4H^+ \rightarrow TiF_6^{2-} + 2H_2O$$

Thus, based on the above results, it can be concluded that fluoride concentration and pH value in the solution affected the stability or solubility of the $TiO_2$ layer on the surface and the corrosion sensitivity of TiNi. Acidic fluoride is prone to destroy the oxide film and reduces the corrosion resistance of TiNi alloy. Furthermore, the preliminary law based on this work will be used for further research on more F components.

### 4.2. Corrosion Comparison between SLM and Wrought TiNi Samples

Above electrochemical corrosion, results show that the SLM TiNi sample presented more superior corrosion resistance in ASS solution containing 0.5% $F^-$ at pH 4, compared with the wrought sample. The reasons may be related to the following two points. Firstly, a large number of inclusions contained in the wrought sample deteriorate the quality of passivation film. It is reported that the corrosion behavior is closely related to the properties of TiNi surface oxide film, including uniformity, density, thickness, etc., [35,36]. Although the wrought TiNi sample has a small grain size, it contains many inclusions such as $Ti_3Ni_4$ precipitate in the grains and along the grain boundaries. These inclusions formed during the forging process and potentially increase the sensitivity of local corrosion [37]. Once the inclusions on the TiNi surface are pulled-out due to the weak inherent bonding, then a lot of pores will be present on the sample surface. It is very difficult for the TiNi sample to form a high-quality passive film on a porous surface [38], leading to accelerated corrosion damage during potentiodynamic polarization. Secondly, the existence of B19′ in the SLM TiNi sample improves its corrosion resistance. Due to the local continuous rapid melting and non-equilibrium solidification during SLM fabrication, the B19′ phase formed in SLM samples, which increased the Ms temperature of TiNi alloy [22]. In addition, the appearance of the B19′ phase may reduce the nickel content on the surface and result in better corrosion resistance of SLM samples [38].

### 5. Conclusions

In this study, the TiNi samples were fabricated by the SLM technique. The effects of microstructure, $F^-$ concentration, and pH value on the corrosion behavior of SLM TiNi

samples in the oral environment were examined and compared with conventional wrought TiNi samples. The main conclusions are as follows:

1. The SLM sample shows slightly better corrosion resistance than that of the wrought sample due to the more uniform and dense passivation film formed on the surface of the SLM sample. The difference in corrosion potential and current density between the two is relatively small. However, the $E_{pit}$ of the former (1.13 V) is significantly higher than that of the latter (0.42 V);

2. The existence of F⁻ (0.2%) in ASS could deteriorate the corrosion resistance of the SLM sample. The $E_{corr}$ of the sample decreases from −0.3 V to −0.36 V, and the $I_{corr}$ increases from $9.85 \times 10^{-2}$ to 13.9 $\mu$A/cm$^2$ due to the addition of fluoride ions. With an increase in F⁻ (0.5%) concentration, the corrosion rate of the SLM sample increased. F⁻ could combine with titanium oxide on the alloy surface to form a titanium–fluorine complex, leading to the passivation film's dissolution;

3. The adjustment of pH value in the fluorine-containing solution may influence the corrosion behavior of SLM samples. The lower the solution's pH value, the worse the alloy's corrosion resistance. As the pH value changes from 5 to 4, the measured $E_{ocp}$ remains the same while $I_{corr}$ (20.8-22.0 $\mu$A/cm$^2$) and $I_{pp}$ (80.1-181 $\mu$A/cm$^2$) increase.

**Author Contributions:** Conceptualization, C.J., S.L., and X.G.; Methodology, C.J., X.W., M.H., Y.S., S.L., X.G., and L.S.; Formal analysis, C.J., S.L., and X.G.; Investigation, C.J. and X.W.; Resources, all materials, devices, and laboratories were provided by the China Institute of Metal Research, Chinese Academy of Sciences; Data Curation, C.J. and X.G.; Writing—Original Draft, C.J.; Writing—Review and Editing, C.J. and S.L.; Supervision, M.H., Y.S., S.L., X.G., and L.S.; Project administration, S.L.; Funding Acquisition, S.L. All authors have read and agreed to the published version of the manuscript.

**Funding:** This work was supported partially by the CAS Interdisciplinary Innovation Team Project (JCTD-2020-10), National Natural Science Foundation of China (51871220, 81902191, 51977132), Natural Science Foundation of Liaoning Province of China (2019-MS-249, 2020-KF-14-01), State Key Laboratory of Light Alloy Casting Technology for High-end Equipment (LACT-007), Shenyang Talents program (RC200230), and the Opening Project of National Key Laboratory of Shock Wave and Detonation Physics (6142A03203002).

**Institutional Review Board Statement:** Not applicable.

**Informed Consent Statement:** Not applicable.

**Data Availability Statement:** Data sharing is not applicable to this article.

**Conflicts of Interest:** The authors declare no conflict of interest.

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
