# Peer review of "Corrosion Behavior of TiNi Alloy Fabricated by Selective Laser Melting in Simulated Saliva"

_coatings, doi:10.3390/coatings12060840_

Round 1

Reviewer 1 Report

Reviewer Recommendation and Comments for manuscript coatings-1729526 with the title: “Corrosion behavior of TiNi alloy fabricated by selective laser melting in simulated saliva”, authors: C. Jia, X. Wang, M. Hu, Y. Su, S. Li, X. Gai, L. Sheng.

The authors present the the influence of microstructure, fluoride ion, and pH value on corrosion behavior of NiTi in saline environment. The alloy was obtained by SLM and by traditional forging technology.

The article may be published after revision.

The main comments that I find useful for improving the quality of the article are presented below:

*The correct template must be used. Line numbers are missing.

* diameter of 10×10×10 mm3

*page 2 - deinontzed water

*Figures 1 to 10 are of poor quality and need to be replaced with higher resolution figures.

*page 3 - ”The oxide layer was unstable after contacting with the corrosive medium and was in a dynamic balance between film formation and dissolution”. What is the scientific basis for this statement? Does the oxide layer not form precisely due to contact with the corrosive environment?

*page 3 – ”trend,reaching 0.38”?

*page 4 – ”accelerate the corrosion of SLM samples”. The OCP method does not provide corrosion rate data.

*Figures 6 and 8 are identical, so only one can be used.

*page 9 – ”passive current densuty”

*page 11 – figure 12 – EDS composition cannot be observed.

*page 12 – please check the stoichiometry of the reactions

*There are many studies on NiTi obtained through SLM. New reference and comparative discussions need to be introduced.

*The typos must be corrected.

*The Coatings journal require a specific format of references, authors must pay more attention in their writing. (eg Abbreviated Journal Name)([CrossRef])

*There are some grammar and typing mistakes.

*The authors must revise the entire manuscript.

Reviewer 2 Report

The work of Jia et al. reports on the formation of TiNi alloys by selective laser melting, compared to the traditional forging method, and evaluates the influence of microstructure, fluoride ion content and pH value on the formed TiNi alloys corrosion behaviour. The following points have to be improved:

  1. Overall, the manuscript needs an English overhaul, to increase the readibility of the text and to correct the existing mistakes. For example: page 1, Introduction: "Saliva is an oral electrolyte can cause electrochemical corrosion of implants";  page 2, Experimental part: "It was optical microscopy (OM) (ZEISS-AXIO), scanning electron microscopy SEM
    (JSM-6510A), and....". etc.
  2. Introduction can be more detailed: for example more details about the SLM techniques, the use of TiNi alloys, more emphasis on the importance of the author's work.
  3. OCP measurements in Fig 3: authors could also include the values for pH 5 without F, and pH 5 with 0.2F - this would result in a more clear picture of the pH influence and fluoride ion content. Similarly for the EIS measurements.
  4. Figure 6 and 8 are identical: though different solution, the equivalent electric circuit is the same in both cases.
  5. Potentiodynamic polarization: a) similar, would be good to include pH 5 without F, and pH 5 with 0.2F; b) others can include in Fig 9, an example on the determination of Icorr and Ipass; c) both Figs 9, 10 font size can be increased
  6. Figs 11, 12 - figure captions need more detail. Fig 12 - b, d - very difficult to read the EDX spectra, authors can replot it or include just the table
  7. Other small things: a) fig 1, XRD: SLM samples - the assignation for the small peaks visible close to 30theta and 65 theta; b) Figure 7 and inset of Fig 7a - increase font size on the axis.

Reviewer 3 Report

Improve the quality of the drawings please. Descriptions of drawings and individual axes are invisible.

Point 3.4. Please start with a capital letter.

There are many words stuck together in the text. Perhaps it is the fault of the software used by me or perhaps it is incorrect editing of the text. Please check the full text of the article.

In summary, please indicate the specific values of the individual factors influencing corrosion not only pure text.

Round 2

Reviewer 1 Report

Reviewer Recommendation and Comments for manuscript coatings-1729526 with the title: “Corrosion behavior of TiNi alloy fabricated by selective laser melting in simulated saliva”, authors: C. Jia, X. Wang, M. Hu, Y. Su, S. Li, X. Gai, L. Sheng.

The authors provided answers to the reviewer comments and obtained a much improved form of the manuscript. I would like to congratulate the authors for their work. I notice some mistakes that the authors need to correct.

*Ti-Ni or TiNi. The same annotation must be used.

*line-70. ”Microstructural characterization was performed by It was optical microscopy”. Please check the grammar.

*line-75. ”The Corrosion morphology”

*line-111. ”The microstructure image of wrought and SLM TiNi samples show in Fig 2.” Please check the grammar.

*line-132. Insert from Figure 3. Please change Chinese to English.

*line-191. Insert from Figure 5b. Please change Chinese to English.

*line-196. Table 1. What is ”Chi-square”?

*line-243. Table 2. What is ”Chi-square”?

*line 259. Tafel or Taffel?

*line-272. Insert from Figure 8. Please change Chinese to English.

*line-196-Table-1 / line-242-Table-2 / line-274-Table-1 / line-321-Table-1. Please check the errors.

*line-347. Figure 11. EDS data must be enlarged

*line-347. Insert from Figure 11-EDS. Please change Chinese to English.

*for ALL references - Abbreviated Journal Name

Reviewer 2 Report

The authors have addressed the points raised by the reviewer.

However, some more minor English corrections or mispelling corrections are needed, see:

- ln 70 "Microstructural characterization was performed by It was optical microscopy (OM)" (remove: it was)

- ln 117: "(a)SLM alloy", space missing after (a)

- legend for purple line in Figure 3, and Figure 8

- previous points 3,5 and the authors's answer: Authors could mention the preliminary law obtained from this work, will be further used for future research with more varied F content.
